# GQ-Net: Training Quantization-Friendly Deep Networks

## Abstract

Network quantization is a model compression and acceleration technique that has become essential to neural network deployment. Most quantization methods perform fine-tuning on a pretrained network, but this sometimes results in a large loss in accuracy compared to the original network. We introduce a new technique to train *quantization-friendly* networks, which can be directly converted to an accurate quantized network without the need for additional fine-tuning. Our technique allows quantizing the weights and activations of all network layers down to 4 bits, achieving high efficiency and facilitating deployment in practical settings. Compared to other fully quantized networks operating at 4 bits, we show substantial improvements in accuracy, for example 66.68% top-1 accuracy on ImageNet using ResNet-18, compared to the previous state-of-the-art accuracy of 61.52% Louizos et al. (2019) and a full precision reference accuracy of 69.76%. We performed a thorough set of experiments to test the efficacy of our method and also conducted ablation studies on different aspects of the method and techniques to improve training stability and accuracy. Our codebase and trained models are available on GitHub.

## 1 Introduction

Neural network quantization is a technique to reduce the size of deep networks and to bypass computationally and energetically expensive floating-point arithmetic operations in favor of efficient integer arithmetic on quantized versions of model weights and activations. Network quantization has been the focus of intensive research in recent years (Rastegari et al., 2016; Zhou et al., 2016; Jacob et al., 2018; Krishnamoorthi, 2018; Jung et al., 2018; Louizos et al., 2019; Nagel et al., 2019; Gong et al., 2019), with most works belonging to one of two categories. The first line of work quantizes parts of the network while leaving a portion of its operations, *e.g.* computations in the first and last network layers in floating point. While such networks can be highly efficient, using bitwidths down to 5 or 4 bits with minimal loss in network accuracy (Zhang et al., 2018; Jung et al., 2018), they may be difficult to deploy in certain practical settings, due to the complexity of extra floating point hardware needed to execute the non-quantized portions of the network. Another line of work aims for ease of real world deployment by quantizing the entire network, including all weights and activations in all convolutional and fully connected layers; we term this scheme *strict quantization*. Maintaining accuracy under strict quantization is considerably more challenging. While nearly lossless 8-bit strictly quantized networks have been proposed (Jacob et al., 2018), to date state-of-the-art 4 bit networks incur large losses in accuracy compared to full precision reference models. For example, the strict 4-bit ResNet-18 model in Louizos et al. (2019) has 61.52% accuracy, compared to 69.76% for the full precision model, while the strict 4-bit MobileNet-v2 model in Krishnamoorthi (2018) has 62.00% accuracy, compared to 71.88% accuracy in full precision.

To understand the difficulty of training accurate low-bitwidth strictly quantized networks, consider a common training procedure which begins with a pre-trained network, quantizes the model, then applies fine-tuning using straight-through estimators (STE) for gradient updates until the model achieves sufficient quantized accuracy. This process faces two problems. First, as the pre-trained model was not initially trained with the task of being subsequently quantized in mind, it may not be "quantization-friendly". That is, the fine-tuning process may need to make substantial changes to the initial model in order to transform it to an accurate quantized model. Second, fine-tuning a model, especially at low bitwidths, is difficult due to the lack of accurate gradient information provided

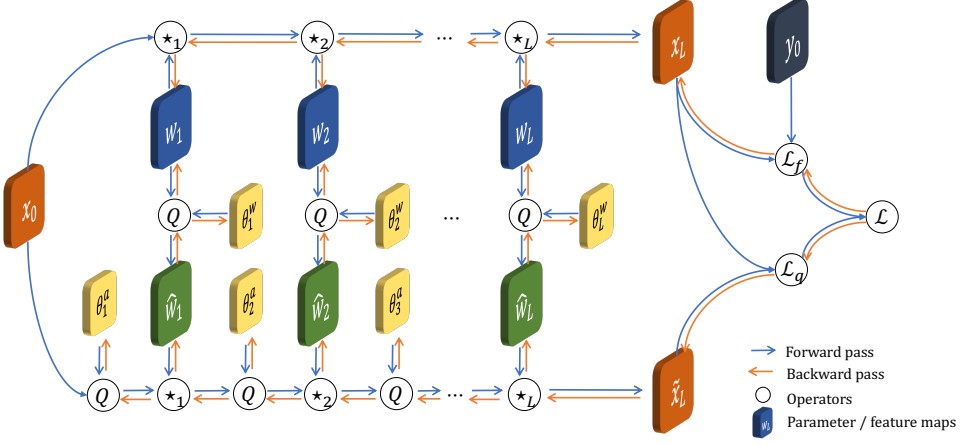

Figure 1: Architecture of the proposed GQ-Net. Input $x_0$ follows the top and bottom paths to produce the full precision and quantized outputs $x_L$ and $\tilde{x}_L$, resp. These are combined through loss functions $\mathcal{L}_f$ and $\mathcal{L}_q$ to form the overall loss $\mathcal{L}$, which is optimized by backpropagation. For more details please refer to Section 3.

by STE. In particular, fine-tuning using STE is done by updating a model represented internally with floating point values using gradients computed at the nearest quantizations of the floating point values. Thus for example, if we apply 4 bit quantization to floating point model parameters in the range $[0, 1]$, a random parameter will incur an average round-off error of $1/32$, which will be incorporated into the error in the STE gradient for this parameter, leading to possibly ineffective fine-tuning.

To address these problems, we propose *GQ-Net*, a *guided* quantization training algorithm. The main goal of GQ-Net is to produce an accurate and quantization-friendly full precision model, *i.e.* a model whose quantized version, obtained by simply rounding each full precision value to its nearest quantized point, has nearly the same accuracy as itself. To do this, we design a loss function for the model which includes two components, one to minimize error with respect to the training labels, and another component to minimize the distributional difference between the model's outputs and the outputs of the model's quantized version. This loss function has the effect of guiding the optimization process towards a model which is both accurate, by virtue of minimizing the first loss component, and which is also similar enough to its quantized version due to minimization of the second component to ensure that the quantized model is also accurate. In addition, because the first component of the loss function deals only with floating point values, it provides accurate gradient information during optimization, in contrast to STE-based optimization which uses biased gradients at rounded points, which further improves the accuracy of the quantized model. Since GQ-Net directly produces a quantized model which does not require further fine-tuning, the number of epochs required to train GQ-Net is substantially less than the total number of epochs needed to train and fine-tune a model using the traditional quantization approach, leading to significantly reduced wall-clock training time. We note that GQ-Net's technique is independent of and can be used in conjunction with other techniques for improving quantization accuracy, as we demonstrate in Section 4.3. Finally, we believe that the guided training technique we propose can also be applied to other neural network structural optimization problems such as network pruning.

We implemented GQ-Net in PyTorch and our codebase and trained models are publicly available [1]. We validated GQ-Net on the ImageNet classification task with the widely used ResNet-18 and

---

[1]An anonymous codebase has been submitted to OpenReview. The GitHub repository will be made public after the review process.

compact MobileNet-v1/v2 models, and also performed a thorough set of ablation experiments to study different aspects of our technique. In terms of quantization accuracy loss compared to reference floating point models, GQ-Net strictly quantized using 4-bit weights and activations surpasses existing state-of-the-art strict methods by up to $2.7\times$, and also improves upon these methods even when they use higher bitwidths. In particular, 4-bit GQ-Net applied to ResNet-18 achieves 66.68% top-1 accuracy, compared to 61.52% accuracy in Louizos et al. (2019) and a reference floating point accuracy of 69.76%, while on MobileNet-v2 GQ-Net achieves 66.15% top-1 accuracy compared to 62.00% accuracy in Krishnamoorthi (2018) and a reference floating point accuracy of 71.88%. Additionally, GQ-Net achieves these results using layer-wise quantization, as opposed to channel-wise quantization in Krishnamoorthi (2018), which further enhances the efficiency and practicality of the technique.

## 2 RELATED WORKS

Neural network quantization has been the subject of extensive investigation in recent years. Quantization can be applied to different part of neural networks, including weights, activations or gradients. Courbariaux et al. (2015), Hou et al. (2016), Zhou et al. (2017), Hou & Kwok (2018) and other works quantized model weights to binary, ternary or multi-bit integers to reduce model size. Wei et al. (2018) quantized activations of object detection models for knowledge transfer. Alistarh et al. (2016), Hou et al. (2019) quantized model gradients to accelerate distributed training. Another line of work quantizes both weights and activations to accelerate model inference by utilizing fix-point or integer arithmetic. These works include Courbariaux et al. (2016), Rastegari et al. (2016), Gysel et al. (2016), Krishnamoorthi (2018), Choi et al. (2018), Zhang et al. (2018), Jung et al. (2018).

A large set of methods have been proposed to improve training or fine-tuning for network quantization. Straight through estimators (Bengio et al., 2013) (STE) propagate gradients through non-differentiable operations with the identity mapping. Other training methods "soften" non-differentiable operations to similar differentiable ones in order for gradients to pass through, then gradually anneal to piecewise continuous functions by applying stronger constraints. This line of works include Louizos et al. (2019), Gong et al. (2019), Bai et al. (2018). Some works regard quantization as a stochastic process that produces parameterized discrete distributions, and guides training using gradients with respect to these parameters Soudry et al. (2014), Shayer et al. (2018). Another line of works does not require fine tuning, and instead re-calibrates or modifies the original network to recover accuracy using little or even no data He & Cheng (2018), Nagel et al. (2019), Meller et al. (2019).

Several recent works have focused on quantizing all parts of a network, typically in order to support deployment using only integer arithmetic units and avoiding the cost and complexity of additional floating point units. Gysel et al. (2016) proposed performing network inference using dynamic fixed-point arithmetic, where bitwidths for the integer and mantissa parts are determined based on a model's weight distribution. Jacob et al. (2018); Krishnamoorthi (2018) proposed the quantization training and deployment algorithm behind the Tensorflow-Lite quantization runtime, which generates strictly quantized networks that can be easily implemented in hardware. Louizos et al. (2019) proposed a training method for strictly quantized models based on annealing a smooth quantization function to a piecewise continuous one. There has also been recent work on using parameterized quantizers which are optimized during quantization training. Choi et al. (2018) introduced learnable upper bounds to control the range of quantization. Zhang et al. (2018) proposed quantizers with a learnable basis which an be executed using fixed-point arithmetic. Jung et al. (2018) proposed to optimize weight scaling and quantization ranges jointly from task losses.

## 3 GQ-NET

In this section we describe the architecture of our proposed GQ-Net and then discuss components of the architecture which can be tuned to improve performance.

### 3.1 GQ-NET ARCHITECTURE

The major components of GQ-Net include the following, and are illustrated in Figure 1:

1. An $L$-layer neural network $h_W(\cdot)$ with all computations performed using full precision floating point arithmetic. Here $W = \{W_1, \ldots, W_L\}$ denotes the parameter (weights) of the model, with $W_i, i \in 1 \ldots L$ being the weights in layer $i$ and expressed in floating point.

2. The quantized model $\hat{h}_{W,Q}(\cdot)$ built from $h_W(\cdot)$. Here $Q = \{Q_1^w, \ldots, Q_L^w, Q_0^a, \ldots, Q_L^a\}$ is a set of *quantizers*, *i.e.* mappings from floating point to (scaled) integer values; the quantizers may be parameterized, and we describe how to optimize these parameters in Section 3.2. $Q_i^w$ quantizes weights $W_i$ and $Q_i^a$ quantizes activations in layer $i$.

   Let $x_0$ denote an input to $h_W$. To construct output $\hat{h}_{W,Q}(x_0)$ of the quantized network, we proceed layer by layer. We first quantize the weights in layers $i = 1, \ldots, L$ as $\hat{w}_i = Q_i^w(w_i)$, and also quantize the input by setting $\hat{x}_0 = Q_0^a(x_0)$. we compute the quantized activations $\hat{x}_i$ in layer $i$ iteratively for $i = 1, \ldots, L$ using $\hat{x}_i = Q_i^a(\tilde{x}_i)$, where $\tilde{x}_i = g_i(\hat{w}_i * \hat{x}_{i-1})$, and $g_i(\cdot)$ denotes the nonlinearity function in layer $i$ and $*$ denotes convolution. Note that since $\hat{w}_i$ and $\hat{x}_{i-1}$ are quantized, $\tilde{x}_i$ can be computed using integer or fixed point arithmetic.

3. Next, we construct a loss function $\mathcal{L}$ incorporating both the training loss $\mathcal{L}_f$ of the full precision model $h_W$ and a loss $\mathcal{L}_q$ capturing the difference between $h_W$ and the quantized model $\hat{h}_{W,Q}$.

$$\mathcal{L} = \omega_f \mathcal{L}_f + \omega_q \mathcal{L}_q \tag{1}$$

   Here $\omega_f, \omega_q \in \mathbb{R}$ are parameters capturing the relative importance of training loss versus distributional loss. In this paper, we focus on image classification networks, and thus we set $\mathcal{L}_f$ to be the cross-entropy loss between outputs from $h_W$ and the training labels. In addition, we set $\mathcal{L}_q = D_{\mathrm{KL}}(\sigma(h_W(\cdot)) \,\|\, \sigma(\hat{h}_{W,Q}(\cdot)))$, where $\sigma$ denotes the softmax function, to be the KL divergence between distributions $\sigma(h_W)$ and $\sigma(\hat{h}_{W,Q})$ on each input. Hence, minimizing the second term in $\mathcal{L}$ corresponds to pushing the floating point and quantized models to behave similarly to each other.

   Since the weight parameters $W$ appear in both terms in $\mathcal{L}$, the two terms can give conflicting signals for updating $W$ during the optimization of $\mathcal{L}$, causing the optimization to be unstable. We discuss how to deal with this problem in Section 3.2.

To train GQ-Net, we successively take mini-batches of training samples and labels and use them to compute $\mathcal{L}$ during the forward pass and propagate gradients with respect to $W$ and the parameters of $Q$ during the backward pass in order to minimize $\mathcal{L}$. After $\mathcal{L}$ has converged sufficiently, we take the quantized weights in $\hat{h}_{W,Q}(\cdot)$ as the quantized model.

## 3.2 Optimizing GQ-Net

We now describe how different components of GQ-Net can be optimized to improve accuracy and training stability.

**Weight scheduling for $\mathcal{L}_f$ and $\mathcal{L}_q$** Parameters $\omega_f$ and $\omega_q$ capture the relative importance of the cross entropy and KL divergence errors during training. A large $\omega_f$ ignores the similarity between the floating point and quantized models and may result in a model that is accurate but not quantization-friendly. Conversely, a large $\omega_q$ ignores guidance on accuracy from the floating point model and may result in similar but poorly performing floating point and quantized models.

We tested different schemes for weighting $\mathcal{L}_f$ and $\mathcal{L}_q$, and found that using fixed values such as $\omega_f = \omega_q = 0.5$ already yields better results than many current methods, as discussed in Section 4. However, further experimentation showed that *scheduling*, *i.e.* dynamically modifying the values of $\omega_f, \omega_q$ during training can produce higher accuracy than using static values. For example, consider a schedule as shown in Figure 2a, which initially sets $\omega_f = 1, \omega_q = 0$, then alternates between setting $\omega_q = 1$ and $\omega_q = 0$ several times. Schedules of this sort can be understood as initially favoring model accuracy so that the floating point model is driven to a high accuracy region of model space, before increasing the importance of model similarity so that a quantization-friendly model can be found in the high accuracy region. This is repeated several times, leading to increasingly more accurate full precision models whose accuracy is then transferred to the quantized model. As demonstrated in Section 4, this schedule results in better performance than static ones.

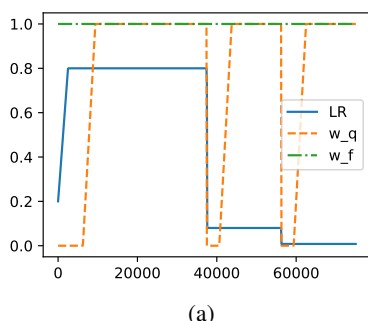 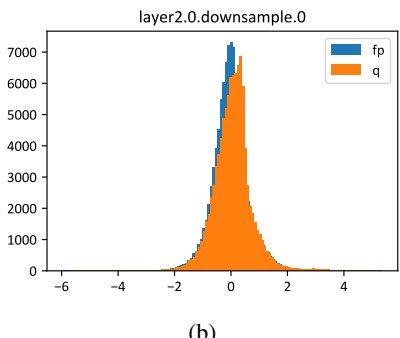

(a)  (b)

Figure 2: (a) Schedule for learning rate, $\omega_f$ and $\omega_q$. $x$-axis shows the number of training steps, $y$-axis shows the value of the learning rate or loss weight. (b) Distribution of `layer2.0.downsample.0` convolution layer outputs in full precision (blue) and 4-bit quantized (orange) ResNet-18, showing the necessity of multi-domain BN. Note that outputs are not quantized until non-linear layers, thus the quantized distribution has more than 16 bins.

**Reducing interference between $\mathcal{L}_f$ and $\mathcal{L}_q$**  The loss function $\mathcal{L}$ includes terms in $\mathcal{L}_f$ and $\mathcal{L}_q$, where the former is a function of the floating point parameters $W$, and the latter involves both $W$ and the quantized version $\hat{W}$ of $W$ parameterized by $\theta$. We discovered that directly optimizing $\mathcal{L}$ results in reduced accuracy, which we attribute to conflicting updates to $W$ produced by gradients from the $\mathcal{L}_f$ and $\mathcal{L}_q$ terms.

Ideally, we would like update $W$ to minimize $\mathcal{L}_f$, while independently updating $\hat{W}$ to minimize $\mathcal{L}_q$. While this is clearly not possible due to the dependency between $W$ and $\hat{W}$, we found that a heuristic based on this idea helped improve accuracy. In particular, let $x_L = h_W(x_0)$ and $\tilde{x}_L = \hat{h}_{W,Q}(x_0)$ be the output of the full precision and quantized networks on input $x_0$, so that $\mathcal{L}_q = D_{\mathrm{KL}}(\sigma(x_L)\,\|\,\sigma(\tilde{x}_L))$, as in Section 3.1. During back propagation we compute $\mathcal{L}_f$ from $x_L$ and derive $\nabla_{x_L}\mathcal{L}_f$ as usual, and use this via the chain rule to update $W$. However, when computing $\mathcal{L}_q$ from $x_L$ and $\tilde{x}_L$, we treat $x_L$ as a constant tensor which does not produce any gradients, and only derive $\nabla_{\tilde{x}_L}\mathcal{L}_q$ and use this via the chain rule to update $W$. We can implement this behavior using the `detach` operator in PyTorch or the `stop_gradient` operator in TensorFlow.

**Parameterized quantizer**  GQ-Net can use any type of quantizer $Q(\cdot) : \mathbb{R} \to \mathbb{T}$ for a discrete set $\mathbb{T}$. Motivated by recent work such as PACT (Choi et al., 2018), we adopt layer-wise linear quantizers with learnable boundaries in GQ-Net. In particular, for each layer $i$, we use one weight quantizer function $Q^w_{i,\theta^w_i}(\cdot)$ for all weights in the layer, and one activation quantizer function $Q^a_{i,\theta^a_i}(\cdot)$ for all activations. Here, $\theta^w_i$ and $\theta^a_i$ represent learnable parameters; for expository simplicity we drop $a, w$ and $i$ in the following and denote all parameters by $\theta$. $\theta$ consists of $k$, the quantization bitwidth, and $lb$ and $ub$ representing the lower and upper quantization boundaries. We use uniform quantization, which is substantially easier to implement in hardware, and set

$$\Delta = \frac{ub - lb}{2^k - 1} \tag{2}$$

$$Q_\theta(x) = \left\lfloor \frac{\mathrm{clamp}(x, lb, ub) - lb}{\Delta} \right\rceil \Delta + lb \tag{3}$$

Where $\mathrm{clamp}(x, a, b) = \max(a, \min(x, b))$, and $\lfloor \cdot \rceil$ represents the round operator.

During training, gradients propagate through the nondifferentiable $\lfloor \cdot \rceil$ operator using the straight though estimator (STE), i.e. $\frac{\partial \lfloor x \rceil}{\partial x} = 1$. Parameters $lb$, $ub$ are updated by the gradients propagated from the loss function $\mathcal{L}$, and thus the quantizers will learn to set the appropriate quantization boundaries to improve accuracy.

**Alternatingly optimizing $W$ and $\theta$**   The accuracy of the quantized model depends both on the weights $W$ of the floating point model as well as how these are quantized using the quantizers parameterized by $\theta$. We found that jointly optimizing $W$ and $\theta$ in each iteration resulted in unstable training and poor accuracy. Performance was substantially improved by alternatingly optimizing $W$ and $\theta$. That is, in each training epoch we update either the values in $W$ or $\theta$ while freezing the values of the other set of parameters, then switch the updated and frozen sets in the next epoch. The reason alternating optimization improved training is that both $W$ and $\theta$ affect the values of quantized weights, so that updating both simultaneously may cause suboptimal changes to quantized weights.

**Multi-domain batch normalization**   Batch normalization is a critical component in large model training. However, since GQ-Net trains a floating point and quantized model simultaneously, we need to adjust the way batch normalization is performed to achieve good accuracy. In particular, as illustrated in Figure 2, activations from the floating point and quantized models follow different distributions. Thus, normalizing them with same set of running statistics can hinder training. Instead, we regard activations in different numerical precision as being from different domains, similar to as in multi-domain transfer learning, and normalize them with separate statistics $\{\mu_f, \sigma_f\}$ and $\{\mu_q, \sigma_q\}$: $\bar{x} = \frac{x - \mu_f}{\sigma_f}$, $\bar{\hat{x}} = \frac{\hat{x} - \mu_q}{\sigma_q}$. This modification only introduces minor storage overhead, while it significantly improves GQ-Net's accuracy in both full-precision and quantized settings.

# 4 EXPERIMENTS

To validate the effectiveness of GQ-Net and assess its different components, we conducted a series of comparisons and ablation studies using the ImageNet classification task. We used the ILSVRC 2012 dataset consists of 1.2M training samples and 10K validation samples from 1K categories, and evaluated our system using top-1 and top-5 validation accuracies.

## 4.1 IMPLEMENTATION DETAILS

**Network settings**   We used the ResNet-18, MobileNet-v1 and MobileNet-v2 architectures in the ImageNet experiments. All MobileNets used channel expansion ratio $1.0$. Unlike some recent works, we did not modify the order of the Conv, BN and ReLU layers. We replaced the BatchNorm layers in these models with SyncBatchNorm, *i.e.* we used mini-batch statistics $\mu_f, \mu_q$ and $\sigma_f, \sigma_q$ computed from all distributed GPUs during training.

**Quantization settings**   Unless otherwise specified, all of the following experiments were conducted with parameterized linear quantizers using a bitwidth of $4$. The weight and activation quantizers each have their own parameters $\theta = \{lb, ub\}$, which are initialized at the iteration right before the quantization error penalty $\mathcal{L}_q$ is enabled. Specifically, for weight quantizers, $\{lb, ub\}$ are initialized by the minimum and maximum elements in the weight tensors of each layer. For activation quantizers, $\{lb, ub\}$ are initialized by the upper and lower $99.9\%$ percentile values, computed from $5$ mini-batches sampled from the training set. Weights and activations of all layers were quantized, including the first and last layers.

**Training protocol**   We used the same training protocol for all architectures and quantization settings. Training was performed on 32 distributed GPUs, each with a mini-batch size of 64, and stopped after 120 epochs on the training set. Model weights $W$ and quantization parameters $\theta$ were optimized with different optimization settings. Model weights $W$ were randomly initialized using the Kaiming-normal scheme (He et al., 2015) without using a pre-trained model, and optimized by SGD with 0.9 Nesterov momentum and $10^{-4}$ weight decay. The learning rate warmed-up from $0.2$ to $0.8$ in the first 4 epochs, and decayed twice by a factor of $0.1$ at epochs 60 and 90. Quantization parameters $\theta$ were optimized using Adam without weight decay, with coefficients for first and second momentums set to $\beta_1 = 0.9$ and $\beta_2 = 0.999$, and learning rate fixed to $10^{-3}$ during the entire training process. Following standard practice, training samples were resized and randomly cropped to $224 \times 224$ pixels, followed by random horizontal flipping and normalization. Validation samples were centrally cropped to $224 \times 224$ pixels, followed by normalization.

Table 1: Comparison with other strict quantization methods on top-1 and top-5 accuracy for ImageNet. W/A indicates the quantization bitwidths for weights and activations, respectively. 32/32 indicates full-precision models.

| Method | W/A | ResNet-18 | | MobileNet-v1 | | MobileNet-v2 | |
|---|---|---|---|---|---|---|---|
| | | Top-1 | Top-5 | Top-1 | Top-5 | Top-1 | Top-5 |
| Reference | 32/32 | 69.76 | 89.08 | 70.60 | | 71.88 | 90.29 |
| Ours | 32/32 | 69.89 | 89.27 | 70.68 | 89.64 | 71.09 | 90.12 |
| White Paper | 4/8 | | | 65.00 | | 62.00 | |
| White Paper | 8/4 | | | 64.00 | | 58.00 | |
| Integer-only | 5/5 | 64.64 | 86.67 | | | | |
| RelaxedQuant | 5/5 | 65.10 | 86.57 | 61.38 | 83.73 | | |
| RelaxedQuant | 4/4 | 61.52 | 83.99 | | | | |
| **GQ-Net** (ours) | 4/4 | 66.68 | 87.46 | 65.04 | 86.09 | 66.15 | 86.92 |

## 4.2 COMPARISON WITH OTHER STRICT QUANTIZATION METHODS

We validated the effectiveness of GQ-Net by comparing it with several other state-of-the-art quantization methods. As GQ-Net seeks to fully quantize a network and execute it using only integer operations, we perform comparisons with other strict quantization methods. The comparison baselines include results from (Krishnamoorthi, 2018), (Jacob et al., 2018) and (Louizos et al., 2019), indicated as White Paper, Integer-only and RelaxedQuant resp. in Table 1.

The first row in the table contains full precision accuracy results evaluated using reference implementations[2], while the second row contains the full precision accuracy for models trained with our training protocol. For the ResNet-18 model which is widely studied in the network compression literature, we significantly outperform the state-of-the-art strict quantization method RelaxedQuant in an equal bitwidth setting (+**5.16**% top-1 accuracy improvement with 4-bit weights and activations). Our 4-bit model even outperforms the comparison methods when they use higher bitwidths (+1.58% compared with 5-bit RelaxedQuant, +2.04% compared with 5-bit Integer-only). For the compact MobileNets family, our method also achieves higher accuracy using lower bitwidths. For example, compared with the White Paper method at 8-bit weights and 4-bit activations, our 4-bit W/A GQ-Net MobileNet-v2 model achieves +**8.15**% top-1 accuracy improvement, and our GQ-Net MobileNet-v1 model obtains +1.04% higher top-1 accuracy in the same setting.

## 4.3 ABLATION STUDIES

We also validated the effectiveness of the different components of GQ-Net discussed in Section 3.2, by progressively adding them to the vanilla quantization-friendly training protocol, and applying these protocols to the ResNet-18 architecture with its weights and activations quantized to 4-bits. For the vanilla setting, $W$ and $\theta$ were optimized jointly at each training step, loss weights were set to $\omega_f = \omega_q = 0.5$ through the entire training process, gradients from $\mathcal{L}_q$ propagated through both $\sigma(x_L)$ and $\sigma(\tilde{x}_L)$, and full precision and quantized activations were normalized using the same set of running statistics. For a more complete comparison, we also evaluated these settings using the full precision GQ-Net model. The numerical results are given in Table 2.

By alternatingly updating model weights $W$ and quantizer parameters $\theta$ between training steps, quantization accuracy improved by +4.24% over the vanilla protocol. This indicates that although gradients $\nabla_W \mathcal{L}$ and $\nabla_\theta \mathcal{L}$ both guide their respective parameters to minimize training loss, combining them in a training step makes the quantized weights $\hat{W}$ derived from these parameters suboptimal.

Dynamically adjusting the weights $\omega_f$ and $\omega_q$ during different parts of the training process as described in Section 3.2 improved quantization accuracy by +0.23%. This suggests that the importance of the accuracy loss $\mathcal{L}_f$ and the distributional loss $\mathcal{L}_q$ may not be equal at different stages of

---

[2]Implementations of ResNet-18 and MobileNet-v2 are taken from `torchvision` package (v0.3.0, `https://pytorch.org/docs/stable/torchvision/models.html`). The accuracy of MobileNet-v1 is cited from the original paper.

Table 2: Full-precision (FP) and quantized (Q) top-1 accuracy for 4-bit weights and activations in ResNet-18 on ImageNet. Components of GQ-Net are progressively applied. The first row indicates the vanilla setting. A check mark in column "Alt $\{W, \theta\}$" means alternatingly optimizing $W$ and $\theta$. A check for "Schedule $\omega_f, \omega_q$" means $\omega_f$ and $\omega_q$ were dynamically adjusted, as described in §3.2. A check for "Detach $\sigma(x_L)$" means that $\sigma(x_L)$ is treated as a constant and we do not propagate $\nabla_{\sigma(x_L)}\mathcal{L}_q$ during the backward pass. A check for "Multi-domain BN" means that full-precision and quantized activations are normalized using separate running statistics. The full-precision reference has top-1 accuracy = 69.89.)

| Alt $\{W, \theta\}$ | Schedule $\omega_f, \omega_q$ | Detach $\sigma(x_L)$ | Multi-domain BN | Top-1 (FP) | Top-1 (Q) |
|---|---|---|---|---|---|
| | | | | 65.21 | 60.95 |
| ✓ | | | | 67.14 | 65.19 |
| ✓ | ✓ | | | 66.56 | 65.42 |
| ✓ | ✓ | ✓ | | 66.61 | 66.08 |
| ✓ | ✓ | ✓ | ✓ | **68.30** | **66.68** |

the training process. The schedule we used, alternating between short periods of $\omega_q = 0$ and longer periods of $\omega_q = 1$, suggests we should first allow the floating point model settle into a reasonably accurate state before enforcing its similarity to the quantized model.

Blocking the gradient from $\mathcal{L}_q$ caused by $\sigma(x_L)$ further improved quantization accuracy by $+0.66\%$. This indicates that although the full precision logits $x_L$ and quantized logits $\tilde{x}_L$ are both derived from the same set of parameters $W$, it is useful to heuristically separate their effects during backpropagation.

Normalizing full precision and quantized activations by different sets of running statistics in the BatchNorm layers improved both full precision accuracy $(+1.69\%)$ and quantized accuracy $(+0.60\%)$. This highlights the difference in the mean and variance of the full precision and quantized activations, despite the similarity of the full precision and quantized models in terms of KL divergence.

Lastly, we considered the effectiveness of using parameterized quantizers. We tested replacing the parameterized quantizers in GQ-Net with naive linear quantizers using fixed weight and activation quantization ranges set to the 99.9% percentile values derived from 5 initial training mini-batches. We found that using fixed quantizers significantly lowered accuracy, by 3.59%. We note that RelaxedQuant also uses learned quantizers, and that replacing these with fixed quantizers may also result in decreased accuracy.

## 5 CONCLUSION

In this paper we presented GQ-Net, a novel method for training accurate quantized neural networks. GQ-Net uses a loss function balancing full precision accuracy as well as similarity between the full precision and quantized models to guide the optimization process. By properly tuning the weights of these two factors, we obtained fully quantized networks whose accuracy significantly exceeds the state of the art. We are currently studying additional ways to adjust GQ-Net components to further improve accuracy. We are also interested in combining GQ-Net with complementary quantization techniques, and in applying similar methodologies to other neural network optimization problems.

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
