# OpenReview forum: "GQ-Net: Training Quantization-Friendly Deep Networks"
_ICLR.cc/2020/Conference — Reject_

### Official Review · AnonReviewer2 · 2019-10-21
**Official Blind Review #2**

**Rating:** 3

**Review:**

This work introduces GQ-Net, a novel technique that trains quantization friendly networks that facilitate for 4 bit weights and activations. This is achieved by introducing a loss function that consists of a linear combination of two components: one that aims to minimize the error of the network on the training labels of the dataset and one that aims to minimize the discrepancy of the model output with respect to the output of the model when the weights and activations are quantized. The authors argue that this has the effect of “guiding” the optimization procedure in finding networks that can be quantized without loss of performance. For the discrepancy metric the authors use the KL divergence from the predictive distribution of the floating point model to the one of the quantized model. The authors then propose several extra techniques that boost the performance of their method: 1. scheduling the weighting coefficients of the two loss terms (something which reminisces iterative pruning methods), 2. stopping the gradient of the floating point model w.r.t. the second loss term, 3. learning the parameters of the uniform quantizer, 4. alternating optimization between the weights and the parameters of the quantizers and 5. using separate batch normalization statistics for the floating point and quantized models. The authors then evaluate their method on Imagenet classification using ResNet-18 and Mobilenet v1 / v2, while also performing an ablation study about the extra tricks that they propose.

This work is well written and in general conveys the main idea in an effective manner. Quantization friendly neural networks in an important subject in order to make deep learning tractable for real world applications. The idea seems on a high level to be interesting and simple; train floating point models that can fit the data well while also encouraging them to be robust to quantization by enforcing the predictive distributions of the fixed and floating point models to be similar in the KL-divergence sense. Nevertheless, I do have some comments that would hopefully help in improving this work:

- It does seem that GQ-Nets need extra tricks in order to perform well, and those tricks come with their own set of hyperparameters that need to be tuned. For example, at section 4.3 you mention that the top-1 accuracy of vanilla GQ-Nets is 60.95, which is lower than the RelaxedQuant baseline (that has 61.52). This raises the question whether the boost in performance is due to the several additional steps employed (which in general can be applied to other quantization techniques as well), and not due to the main idea itself.
- Do you employ the straight-through estimator (STE) for the weights in the L_q objective? In the second paragraph of the second page you argue that due to the biased gradients of STE the performance is in general reduced, so I was wondering whether STE posed an issue there or whether you used an alternative estimator.
- How is batch normalization handled? Do you absorb the scale and shifts in the weights / biases before you perform quantization or do you quantize the weights and then apply the BN scale and shift in full precision?
- How do you ensure and ub > lb when you learn the quantizer? In general learning the quantizer can be also done with alternative techniques (e.g. simply learning the scale and offset) so I was wondering whether you noticed benefits from using the ub, lb parametrization compared to others.
- Do you show the pre-quantization distributions at Figure 2b? In the caption you mention quantized but the resolution seems to be higher than the 16 values you should get with 4 bits. Furthermore, it should be noted that the discrepancy in BN in quantized models was, as far as I am aware, firstly noticed at [1] (and subsequently at RelaxedQuant) and both of these methods simply re-estimated the moving averages during the inference time.

Overall, I am on the fence about this work and tend to reject. Having said that, I am of course willing to revise my score after the discussions with the authors / other reviewers.

[1] Probabilistic Binary Neural Networks, Jorn W.T. Peters, Max Welling

**Experience Assessment:**

I have published one or two papers in this area.

**Review Assessment: Checking Correctness Of Derivations And Theory:**

I assessed the sensibility of the derivations and theory.

**Review Assessment: Checking Correctness Of Experiments:**

I assessed the sensibility of the experiments.

**Review Assessment: Thoroughness In Paper Reading:**

I read the paper thoroughly.

---

> ### Author Response · Authors · 2019-11-15
> **Response to Review #2 (1/2)**
>
> Thank you for your detailed review and comments. Following are our renposes for each of your concerns:
>
> Q1: Whether the boost in performance is due to the several additional steps employed, and not due to the main idea itself?
>
> A: Thank you for your detailed review and comments.  As Review #1 made a similar comment, our response here is similar to our response for Review #1.
>
> We argue that several of the techniques described in Section 3 are part of GQ-Net quantization framework itself, and not orthogonal heuristics which are added to improve performance.  In particular, we show that weight scheduling, detaching gradients and multidomain batch normalization arise naturally and in a principled way when optimizing GQ-Net.  We also justify the use of alternating training and learned quantizers, and describe a setting where they are not needed.
>
> - Multidomain BN: GQ-Net essentially optimizes two models at the same time, namely the full precision and quantized models. Since these models have substantially different statistics, we use different BN moving averages for each.  This approach parallels traditional fine-tuning based quantization, where the pre-trained full precision model uses one set of BN moving averages, and the fine-tuned quantized model uses a different set of BN values produced on the basis of the first.
>
> - Dynamic weight scheduling: GQ-Net tries to find a model that is both accurate and easily quantizable.  However, during training it must optimize the first objective before the second, as otherwise it may produce a floating point model which is similar to its quantized version, but where both models are inaccurate.  To prioritize initially for accuracy, we can use a simple schedule where both objectives are weighted equally.  Since the accuracy loss and gradients are both initially larger than the quantizability loss and gradients, this schedule has the effect of prioritizing for accuracy.  We can prioritize accuracy further by removing the quantizability loss for a few epochs at the start of training and also each time we change the learning rate.  At these time points, as commonly observed, the full precision model has the opportunity to significantly improve its accuracy, and so the schedule focuses on this objective while temporarily ignoring quantizability.
> We can also try to produce a good weight schedule automatically.  For example, during training we can dynamically set the weights to equalize the two loss terms in Equation 1.  We will study automatic weight scheduling more in future work.
>
> - Detached gradients: This helps reduce interference between the accuracy and quantizability losses.  If gradients from the quantizability loss directly propagate to the weights (i.e. if there was an orange arrow from $\mathcal{L}_q$ to $x_L$ in Figure 1), this may lead to weight changes which improve quantizability but decrease accuracy.  Detaching gradients somewhat reduces this effect, and encourages weights and theta parameters to change to improve quantizability while maintaining accuracy.
>
> - Alternating training for W and theta: Since both W and theta affect quantization, training them jointly may cause interference.   It may be possible to train W and theta jointly by using different learning rates for each set of parameters (as done in some works on trained quantizers, e.g. PACT [1] and LIQ [2]), but this requires a careful selection of learning rates.  We found alternating training to be simpler and equally or more effective.
>
> - Learned quantizers:  Learned quantizers are orthogonal to the main idea of GQ-Net.  We tested its importance to GQ-Net by removing it, while still using multidomain BN, dynamic weight scheduling and detached gradients, which we argued above were core components of GQ-Net.  Our accuracy decreased by 3.59% to 63.09%.  However, this still exceeds RelaxedQuant’s accuracy of 61.52% in the 4/4 configuration.  Furthermore, we note that RelaxedQuant itself uses learned quantization.
>
> In addition, we performed a 5 bit quantization experiment without learned quantizers (or alternating training), but using multidomain BN, weight scheduling and detached gradients.  This achieved 67.6% accuracy, which is higher than Integer-only (64.64%) and RelaxedQuant’s accuracy (65.1%) in the 5/5 setting.
>
> [1] PACT: Parameterized clipping activation for quantized neural networks, Choi et al.
> [2] Learning to Quantize Deep Networks by Optimizing Quantization Intervals with Task Loss, Jung et al.

---

> ### Author Response · Authors · 2019-11-15
> **Response to Review #2 (2/2)**
>
> Q2: Do you employ the straight-through estimator (STE) for the weights in the $\mathcal{L}_q$ objective?
>
> A: Yes, we use STE for the $\mathcal{L}_q$ objective, and so our gradients are biased when optimizing $\mathcal{L}_q$.  However, GQ-Net’s loss comes from both the $\mathcal{L}_q$ and $\mathcal{L}_f$ (floating poing accuracy) objectives. While gradients for $\mathcal{L}_q$ are biased, gradients for $\mathcal{L}_f$ are computed accurately in floating point.  By contrast, in the fine tuning stage of traditional quantization methods the gradients for the quantized model are computed entirely using STE. Thus, overall GQ-Net uses more accurate gradients, which is one of the reasons for its improved accuracy.
>
> Q3: How is batch normalization handled?
>
> A: Our current implementation quantizes the weights and then applies the shift and scaling terms in full precision.  However, as we did not change the order of the model operators, the BN terms can be absorbed into the weights of the previous Conv / FC layer before model deployment.  So at inference time we can use only fixed point operations.
>
> Q4: How do you ensure and ub > lb when you learn the quantizer? Is this better than learning quantization scale and bias terms?
>
> A: We did not explicitly enforce ub > lb during the training process, but in practice we found that the condition was never violated.  One reason for this is that the lb and ub values are initially widely separated, and then maintained separation throughout training.  Specifically, we initialized lb and ub for the weights to be the minimum and maximum random initial weight values.  For activations, we ran the initial minibatch of inputs through the network in full precision, then set lb and ub to be the 99.9 percentile minimum and maximum activation values we observed.
>
> In general, learning quantization boundaries is equivalent to learning scaling and bias terms, since these terms can be directly derived from boundaries, and the mappings between them are linear, so this will not introduce any additional learning difficulty. We choose to learn quantization boundaries due to our implementation choices.
>
> Q5: About the distribution illustrated in Figure 2b?
>
> A: Thank you for pointing this out.  Figure 2b is correct, but the caption is unclear.  The orange values in Figure 2b are the outputs after convolution with 4-bits quantized weights and input in a particular layer (we arbitrarily chose layer layer2.0.downsample.0 in ResNet-18, but all layers behave similarly).  These values will then be passed through the batch norm and the nonlinearity function of the layer before finally being quantized, thus the histogram has more than 16 bins. We have revised the caption in Figure 2b to make this clear.

---

> ### Comment · AnonReviewer2 · 2019-11-15
> **Thank you for the detailed response**
>
> I appreciate the detailed reply of the authors. Overall, the new results make me feel more positive about this work, but I decided to keep my rating the same. This is primarily for two reasons; While I can understand that some of the additional steps can be understood as part of the main method (eg detaching the gradients), other steps do seem orthogonal. The weight scheduling for example can be employed even for STE type of approaches, as one can similarly gradually apply the quantization operator. The method overall seems to have quite a few moving moving parts and I am not entirely sure how robust it is to the hyper parameters / schedules.
>
> Furthermore, the authors argued that the gradients are more accurate due to having the extra loss term for the floating point model. In general this is not true, as at the end of the day we care about the quantized model and not the full precision one (as R3 also pointed out). Finally,  one should be careful in the interpretation of the results with unquantized batchnorm; absorbing the scales and offsets changes the marginal distribution of the weights and biases, hence can introduce errors to the subsequent quantizer (as the model was optimized under a specific distribution).

---

### Official Review · AnonReviewer3 · 2019-10-24
**Official Blind Review #3**

**Rating:** 3

**Review:**

The paper propose a new quantization-friendly network training algorithm called GQ (or DQ) net. I addresses the existing issues in the common paradigm, where a floating-point network is trained first, followed by a second-phase training step for the quantized version. It is a well-written paper. Concepts were clearly explained and easy to follow. Below I present my comments about some details in the paper that were not entirely clear for me.

- The two loss terms conflict each other. If the training algorithm focuses too much on the first term, it will make the network less friendly to the quantization process. On the other hand, the second one is going to enforce too much emphasis on the accuracy from the quantized network. It is natural to involve some hyperparameter search to find the balance between the two blending parameters. The paper suggests a strategy as to how to handle this issue, but it is not comprehensive, and rather controversial. I think the paper will benefit from a more in-depth discussion and analysis on this regularization issue.

- The schedule for the loss term blending parameters looks drastic to me. It’s more like “train the floating point net first, and then train the quantized one, and then revisit the floating point one, and so on.” I know I simplified, because the floating point network never stops getting updated as it’s \omega_f is always 1. However, it seems to me that this drastic scheduling strategy sounds like very similar to the traditional approach that trains the floating point network first and then finetune the quantized one, except for the fact that this proposed algorithm repeats this process a few times. Hence, I think the authors’ argument about the supremacy of the proposed method to the two-step finetuning approach is not clearly supported.

- The exponentially decaying learning rate scheduling looks like the one from ResNet. I’m wondering if it should be the best, especially with the drastic introduction and omission of the second loss.

- In the ablation studies, it seems that some of the suggested training options are conflicting each other and the clear winner seems to be the multi-domain BN. I cannot conclude anything from this analysis as to which one is more important than the other one, except for the Alt{W,\theta} case.

Some minor things:

- What’s the name of the proposed network? Is it GQ or DQ?


**Experience Assessment:**

I have published one or two papers in this area.

**Review Assessment: Checking Correctness Of Derivations And Theory:**

I carefully checked the derivations and theory.

**Review Assessment: Checking Correctness Of Experiments:**

I carefully checked the experiments.

**Review Assessment: Thoroughness In Paper Reading:**

I read the paper at least twice and used my best judgement in assessing the paper.

---

> ### Author Response · Authors · 2019-11-15
> **Response to Review #3 (1/2)**
>
> Thank you for your detailed review and comments. Following are our renposes to each of your concerns:
>
> Q1: The two loss terms conflict each other. The paper will benefit from a more in-depth discussion and analysis on this regularization issue.
>
> A: Thank you for your detailed review and comments.  We agree the two loss terms may conflict with each other, and used several techniques to reduce this effect.  For example, we adjust weights for the accuracy and quantizability loss terms in Equation 1 so that the training process will initially focus on finding an accurate model before optimizing for the model’s quantizability.  This is important because otherwise GQ-Net may produce a floating point model which is similar to its quantized version, but where both models are inaccurate.  One schedule we used set the weights for the two terms to be equal.  Since the accuracy loss and gradients are both initially larger than the quantizability loss and gradients, this schedule has the effect of prioritizing for accuracy.  We could also improve accuracy somewhat (+0.23%) by using a schedule which set the quantizability loss weight to 0 for a few epochs at the start of training and also each time we reduced the learning rate.  At these time points, as commonly observed, the full precision model has the opportunity to significantly improve its accuracy, and so the schedule focuses on this objective while temporarily ignoring quantizability.
>
> Additionally, we can try to produce a good weight schedule automatically.  For example, during training we can dynamically set the weights to equalize the two loss terms in Equation 1.  We will study automatic weight scheduling more in future work.
>
> Another way we reduce conflict between the two losses is by detaching gradients, so that the gradient from $\mathcal{L}_q$ is not propagated to $x_L$ (this is illustrated by the lack of an orange arrow from $\mathcal{L}_q$ to $x_L$ in Figure 1).  We do this to prevent the model from trying to reduce its overall loss by modifying weights to reduce the quantizability loss while increasing the accuracy loss.  Using detached gradients, the network is more likely to modify weight and theta parameters to reduce quantizability loss while preserving accuracy loss.  As shown in our ablation study, this improves accuracy by 0.66%.
>
> Q2: The schedule for the loss term blending parameters looks drastic. What is the benifit compreing with two-step quantization finetune?
>
> A: Our schedule only tunes the weights for a few epochs at the start of training, and when learning rates are lowered at epochs 60 and 90.  There is also an important difference between GQ-Net and alternating training between a floating point model and a quantized one, namely that GQ-Net uses more accurate gradient information during training than traditional fine tuning of a quantized model.
>
> In particular, GQ-Net’s gradient comes from the two terms in Equation 1.  The first term’s gradient is computed accurately using floating point, while the second term’s gradient is biased because it is computed at quantized points using straight through estimators (STE).  For traditional methods, during fine-tuning of the quantized model the gradient is computed entirely using STE, and thus incurs more bias.
>
> Q3: About the learning rate schedule in the experiments?
>
> A: We used a standard learning rate schedule for ResNet-18, and also tied the weight schedule for the accuracy and quantizability losses to the learning rate schedule to let the model optimize first for accuracy and then for quantizability, as described above.  We experimented with some other learning rate schedules, but found this schedule resulted in the most stable training and also good accuracy.

---

> ### Author Response · Authors · 2019-11-15
> **Response to Review #3 (2/2)**
>
> Q4: About the interpretation of ablation study?
>
> A: Our ablation study indicated that multidomain BN has a moderate impact on accuracy (+0.6%), due to the significantly different activation statistics between the full precision and quantized models.  Detaching gradient also had a moderate effect (0.66%) because it helped reduce interference between the accuracy and quantizability losses, while scheduling weights had a more limited effect (+0.23%).
>
> The effectiveness of alternating training and using a learned quantizer is context dependent.  For example, for 4 bit quantization both techniques had a significant impact (+4.24% and +3.59% resp).  When we removed the learned quantizer (and hence also remove alternating training of the learned quantizer’s parameters theta), GQ-Net has a 4 bit accuracy of 63.09%.  We note that this still exceeds RelaxedQuant’s accuracy of 61.52% in the 4/4 configuration, and that RelaxedQuant itself uses a learned quantizer.
>
> We also found that for 5 bit quantization, we could obtain state of the art results without using either alternating training or learned quantizers.  Specifically, we used ResNet-18 and Imagenet, and performed strict 5 bit quantization of all weights and activations, and used only weight scheduling, detached gradients and multidomain BN.  We obtained 67.6% top-1 accuracy, compared to the Integer-only and RelaxedQuant models which achieve 64.64% and 65.1% accuracy resp. at the 5/5 setting.
>
> Q5: What’s the name of the proposed network?
>
> A: Thank you for pointing this out.  We call our network GQ-Net, for guided quantization.  We have fixed the typo throughout the paper.

---

> ### Comment · AnonReviewer3 · 2019-11-15
> **Thanks for the response**
>
> I appreciate the authors' responses that clarified some of my questions. The responses elaborated the arguments made in the original draft, while they do not fully resolve the fundamental issues. For example, the I wouldn't say the gradient from the first loss term is more accurate, as it's using full precision, which is "different" from the test environment where reduced precision has to be used. They also suggest a few potential solutions, while the revised version doesn't really contain those ideas.

---

### Official Review · AnonReviewer1 · 2019-10-26
**Official Blind Review #1**

**Rating:** 6

**Review:**

In this paper, the authors propose a framework towards 4-bit auantization of CNNs. Specifically, during training, the proposed method contains a full precision branch supervised by classification loss for accurate prediction and representation learning, as well as a parameterized quantization branch to approximate the full precision branch. A quantization loss between the full precision branch and the quantization branch is defined to minimize the difference between activation distributions. The authors proposed a series of improvements, including alternative optimization, dynamic scheduling, detach and batch normalization to help boosting the performance to SOTA under 4-bit quantization.

Strengths:
+ Well-written paper with good clarity and technical correctness.
+ Proposed method seems light, sweet and technically correct.
+ Good experimental performance and result on ImageNet.
+ Good and clear ablation study.

Weaknesses:
- Major performance improvement comes from the combination of different incremental improvements.
- Lack of evaluations with variety of datasets (CIFAR-10/MNIST)/configurations (other bitwidth)
- Lack of the citation and comparison to many most recent works on binarized networks (except XNOR-Net)

Comments:
I consider this a well-written paper with great clarity and good empirical performance. I enjoyed reading the paper. The proposed framework seems technically correct and effective.

However, a major weakness of this work is that most of the performance improvement comes from a combination of add-on improvements, except that the authors put them together into a unified framework and explained elegantly. The vanilla architecture, which is a main contribution and described in Fig. 1, doesn't seem to give that significant improvement. To some extent, the real technical contributions of this work are partly weakened given the add-on combinations and the existence of similar methods. For example, the alternative optimization of W and \theta is similar to alternative re-training in network pruning, although a unified loss/optimization framework is applicable in this case. Others such as dynamic scheduling and gradient detach are also heuristic-driven.

The results on ImageNet under 4-bit quantization are strong and convincing, but the paper could benefit from conducting additional experiments on different datasets and bitwidth configurations. A more comprehensive study similar to Louizos et al., 2019 will be great. Citations and comparisons to more recent binarized networks other than XNOR-Net will be appreciated too.

**Experience Assessment:**

I have read many papers in this area.

**Review Assessment: Checking Correctness Of Derivations And Theory:**

I carefully checked the derivations and theory.

**Review Assessment: Checking Correctness Of Experiments:**

I assessed the sensibility of the experiments.

**Review Assessment: Thoroughness In Paper Reading:**

I read the paper at least twice and used my best judgement in assessing the paper.

---

> ### Author Response · Authors · 2019-11-15
> **Response to Review #1 (1/2)**
>
> Thank you for your detailed review and comments. Following are our renposes to each of your concerns:
>
> Q1: Most of the performance improvement comes from a combination of add-on improvements, contribution from the main idea is thus weaken?
>
> A: Thank you for your detailed review and comments.  Regarding the optimizations in Section 3, we argue that several of these techniques are part of GQ-Net quantization framework itself, and not orthogonal heuristics which are added to improve performance.  In particular, we show that weight scheduling, detaching gradients and multidomain batch normalization arise naturally and in a principled way when optimizing GQ-Net.  We also justify the use of alternating training and learned quantizers, and describe a setting where they are not needed.
>
> - Multidomain BN:  GQ-Net essentially optimizes two models at the same time, namely the full precision and quantized models. Since these models have substantially different statistics, we use different BN moving averages for each.  This approach parallels traditional fine-tuning based quantization, where the pre-trained full precision model uses one set of BN moving averages, and the fine-tuned quantized model uses a different set of BN values produced on the basis of the first.
>
> - Dynamic weight scheduling:   GQ-Net tries to find a model that is both accurate and easily quantizable.  However, during training it must optimize the first objective before the second, as otherwise it may produce a floating point model which is similar to its quantized version, but where both models are inaccurate.   To prioritize initially for accuracy, we can use a simple schedule where both objectives are weighted equally.  Since the accuracy loss and gradients are both initially larger than the quantizability loss and gradients, this schedule has the effect of prioritizing for accuracy.  We can prioritize accuracy further by removing the quantizability loss for a few epochs at the start of training and also each time we change the learning rate.  At these time points, as commonly observed, the full precision model has the opportunity to significantly improve its accuracy, and so the schedule focuses on this objective while temporarily ignoring quantizability.
> We can also try to produce a good weight schedule automatically.  For example, during training we can dynamically set the weights to equalize the two loss terms in Equation 1.  We will study automatic weight scheduling more in future work.
>
> - Detached gradients:  This helps reduce interference between the accuracy and quantizability losses.  If gradients from the quantizability loss directly propagate to the weights (i.e. if there was an orange arrow from $\mathcal{L}_q$ to $x_L$ in Figure 1), this may lead to weight changes which improve quantizability but decrease accuracy.  Detaching gradients somewhat reduces this effect, and encourages weights and theta parameters to change to improve quantizability while maintaining accuracy.
>
> - Alternating training for W and theta:  Since both W and theta affect quantization, training them jointly may cause interference.   It may be possible to train W and theta jointly by using different learning rates for each set of parameters (as done in some works on trained quantizers, eg PACT [1] and LIQ [2]), but this requires a careful selection of learning rates.  We found alternating training to be simpler and equally or more effective.
>
> - Learned quantizers:  Learned quantizers are orthogonal to the main idea of GQ-Net.  We tested its importance to GQ-Net by removing it, while still using multidomain BN, dynamic weight scheduling and detached gradients, which we argued above were core components of GQ-Net.  Our accuracy decreased by 3.59% to 63.09%.  However, this still exceeds RelaxedQuant’s accuracy of 61.52% in the 4/4 configuration.  Furthermore, we note that RelaxedQuant itself uses learned quantization.
>
> In addition, we performed a 5 bit quantization experiment without learned quantizers (or alternating training), but using multidomain BN, weight scheduling and detached gradients.  This achieved 67.6% accuracy, which is higher than Integer-only (64.64%) and RelaxedQuant’s accuracy (65.1%) in the 5/5 setting.
>
> [1] PACT: Parameterized clipping activation for quantized neural networks, Choi et al.
> [2] Learning to Quantize Deep Networks by Optimizing Quantization Intervals with Task Loss, Jung et al.

---

> ### Author Response · Authors · 2019-11-15
> **Response to Review #1 (2/2)**
>
> Q2:  The paper could benefit from conducting additional experiments on different datasets and bitwidth configurations.
>
> A: Thank you for the suggestion.  We conducted an additional experiment on 5 bit quantization, as described above.  The experiment used ResNet-18 and Imagenet, as we believe this model and dataset are more similar to practical deployment scenarios.  In detail, we performed strict 5 bit quantization of all weights and activations, and used the weight scheduling (same schedule as in shown in Figure 2a), detached gradients and multidomain BN, but did not used learned quantizers or alternating training. We achieved 67.6% accuracy, compared to Integer-only (64.64%) and RelaxedQuant (65.1%) in the 5 bit weight and activation setting.
>
> Q3: Citations and comparisons to more recent binarized networks other than XNOR-Net.
>
> A: We compare to GQ-Net to two recent binarized networks, DoReFa-Net [3] and HWGQ [4], both in the 1 bit  weights / 4 bit activations configuration.  DoReFa-Net has an accuracy of 59.2%, and HWGQ has an accuracy of 59.6% using the ResNet-18 architecture on Imagenet.  We did not test our network in the 1/4 configuration, and we believe it may not perform as well as DoReFa-Net and HWGQ in this very stringent regime. However, we note that DoReFa-Net performs the first and last layers in floating point, whereas our network is fully quantized and thus runs using only fixed point hardware.  Also, HWGQ also performs the first and last layers in floating point, and furthermore uses nonuniform quantization, which requires dedicated hardware to run.
>
> [3] DoReFa-Net: Training Low Bitwidth Convolutional Neural Networks with Low Bitwidth Gradients, Zhou et al.
> [4] Deep Learning with Low Precision by Half-wave Gaussian Quantization, Cai et al.

---

### Decision · Program_Chairs · 2019-12-19

**Decision:**

Reject

**Comment:**

The paper propose a new quantization-friendly network training algorithm called GQ (or DQ) net. The paper is well-written, and the proposed idea is interesting. Empirical results are also good. However, the major performance improvement comes from the combination of different incremental improvements. Some of these additional steps do seem orthogonal to the proposed idea. Also, it is not clear how robust the method is to the various hyperparameters / schedules. For example, it seems that some of the suggested training options are conflicting each other. More in-depth discussions and analysis on the setting of the regularization parameter and schedule for the loss term blending parameters will be useful.